

# Metabolic syndrome in hospitalized patients with chronic obstructive pulmonary disease

Evgeni Mekov[1], Yanina Slavova[1], Adelina Tsakova[2], Marianka Genova[2], Dimitar Kostadinov[1], Delcho Minchev[1] and Dora Marinova[1]

[1] Clinical Center for Pulmonary Diseases, Medical University—Sofia, Sofia, Bulgaria
[2] Central Clinical Laboratory, Medical University—Sofia, Sofia, Bulgaria

Corresponding author
Evgeni Mekov, dr_mekov@abv.bg

## ABSTRACT

**Introduction.** The metabolic syndrome (MS) affects 21–53% of patients with chronic obstructive pulmonary disease (COPD) with a higher prevalence in the early stages of COPD, with results being highly variable between studies. MS may also affect natural course of COPD—number of exacerbations, quality of life and lung function.

**Aim.** To examine the prevalence of MS and its correlation with comorbidities and COPD characteristics in patients with COPD admitted for exacerbation.

**Material and methods.** 152 patients with COPD admitted for exacerbation were studied for presence of MS. All of them were also assessed for vitamin D status and diabetes mellitus type 2 (DM). Data were gathered for smoking status and exacerbations during the last year. All patients completed CAT (COPD assessment test) and mMRC (Modified Medical Research Council Dyspnea scale) questionnaires and underwent spirometry. Duration of current hospital stay was recorded.

**Results.** 25% of patients have MS. 23.1% of the male and 29.5% of the female patients have MS ($p > 0.05$). The prevalence of MS in this study is significantly lower when compared to a national representative study (44.6% in subjects over 45 years). 69.1% of all patients and 97.4% from MS patients have arterial hypertension. The presence of MS is associated with significantly worse cough and sleep (1st and 7th CAT questions; $p = 0.002$ and $p = 0.001$ respectively) and higher total CAT score ($p = 0.017$). Average BMI is 27.31. None of the patients have MS and BMI $<25$. There is a correlation between the presence of MS and DM ($p = 0.008$) and with the number of exacerbations in the last year ($p = 0.015$). There is no correlation between the presence of MS and the pulmonary function.

**Conclusion.** This study among hospitalized COPD patients finds comparable but relatively low prevalence of MS (25%) compared to previously published data (21–53%) and lower prevalence compared to general population (44.6%). MS may impact quality of life and the number of exacerbations of COPD. Having in mind that MS is more common in the early stages and decreases with COPD progression, the COPD patients admitted for exacerbation may be considered as having advanced COPD.

## INTRODUCTION

Chronic Obstructive Pulmonary Disease (COPD) is a preventable and treatable disease with significant extrapulmonary effects that may contribute to the severity in individual patients. By 2030, COPD will be the fourth cause of mortality worldwide. The extrapulmonary comorbidities influence the prognosis of the patients with COPD (*Global Initiative for Chronic Obstructive Lung Disease , 2014*).

Metabolic syndrome (MS) is common in patients with COPD. According to the available studies the prevalence of MS in COPD patients varies between 21–53% (*Mekov & Slavova, 2013*). The prevalence of MS in COPD patients is increased when compared to a control group (*Funakoshi et al., 2010*; *Lam et al., 2010*; *Marquis et al., 2002*; *Park et al., 2012*).

Available studies suggest that MS may have impact on quality of life (*Ford & Li, 2008*), lung function (*Fimognari et al., 2007*; *Leone et al., 2009*; *Lin et al., 2006*; *Nakajima et al., 2008*; *Yeh et al., 2011*), natural course of COPD (number of exacerbations) (*Kupeli et al., 2010*; *Abdelghaffar et al., 2012*) as well as to affect comorbidities in COPD patients (*Mekov & Slavova, 2014*).

Many studies examine prevalence of MS in COPD patients (*Funakoshi et al., 2010*; *Lam et al., 2010*; *Park et al., 2012*; *Kupeli et al., 2010*; *Akpinar et al., 2012*; *Marquis et al., 2005*; *Minas et al., 2011*; *Ozgen Alpaydin et al., 2013*; *Poulain et al., 2008*; *Watz et al., 2009*) with results being highly variable between studies. The prevalence of MS in an unselected Bulgarian population aged 20–80 years is 30.8%. The prevalence of MS for participants over 45 years (the most common age group for the COPD patients) is higher, 44.6% (*Borissova et al., 2007*). An epidemiological study conducted in Bulgaria in 3,598 COPD patients showed that metabolic syndrome is found in 13.8% of the patients (*Pavlov et al., 2010*). A more recent study indicates that the prevalence of metabolic syndrome is 41.8% in 141 patients with COPD compared to 39% in the control group of 103 subjects (*Stratev et al., 2012*). The prevalence of MS in COPD patients, hospitalized for exacerbation is hard to predict because MS tends to be more prevalent in early stages of COPD while patients experiencing severe exacerbation often have advanced disease. On the other side, MS may impact natural course of COPD and predispose to exacerbation which will lead to increased prevalence of MS in this group.

There is not enough data to determine whether the results from these studies are applicable to specific subgroups of patients, such as COPD patients admitted for exacerbation. COPD is increasingly divided in subgroups or phenotypes based on specific features and association with prognosis or response to therapy, the most notable being the feature of frequent exacerbations (*Vestbo, 2014*). The presence of MS may also have distinctive characteristics for this subgroup ('severe' exacerbator phenotype). The aim of this study is to find out the prevalence of MS in patients with COPD admitted for exacerbation and the correlations of presence of MS with comorbidities and COPD characteristics.

## MATERIAL AND METHODS

A total of 152 COPD patients hospitalized for exacerbation were studied for the presence of MS, DM, and vitamin D deficiency and insufficiency using well-established criteria for:

- Presence of MS: at least 3 of the following: 1. Elevated waist circumference >102 cm in males, >88 cm in females; 2. Triglycerides >1.7 mmol/L (or on therapy); 3. HDL <1.0 mmol/L in males, <1.3 mmol/L in females (or on therapy); 4. Elevated blood pressure: systolic ≥130 and/or diastolic ≥85 mm Hg (or on therapy); 5. Fasting glucose >5.5 mmol/L (or on therapy) (*Alberti et al., 2009*).
- Presence of DM: fasting plasma glucose ≥7.0 mmol/L OR 2-h plasma glucose ≥11.1 mmol/L during an oral glucose tolerance test (OGTT) OR HbA1c≥6.5% OR on therapy (*American Diabetes Association, 2012*);
- Presence of prediabetes: fasting plasma glucose 5.6–6.9 mmol/L OR 2-h plasma glucose 7.8–11.0 mmol/L during an OGTT OR HbA1c 5.7–6.4% (*American Diabetes Association, 2012*);
- Presence of vitamin D deficiency: 25(OH)D <25 nmol/L; vitamin D insufficiency: 25(OH)D 25–50 nmol/L; vitamin D sufficiency: >50 nmol/L (*Borissova et al., 2012*).

The diagnosis of COPD was made according to GOLD (Global Initiative for Chronic Obstructive Lung Disease) criteria (DM1). Data were gathered for age, sex, smoking status and number of pack-years, number of bone fractures, therapy for arterial hypertension, therapy for DM, COPD therapy and number of exacerbations in the last year. The patients completed CAT and mMRC questionnaires and underwent pre- and post bronchodilatatory spirometry. Blood pressure was obtained according to the American Heart Association Guidelines (*Pickering et al., 2005*). A patient was considered as having arterial hypertension if taking antihypertensives.

The inclusion criteria were post bronchodilator spirometry obstruction defined as FEV1/FVC<0.70. All participants in this study signed informed consent.

The exclusion criteria were failure to comply with study procedures (no completed questionnaires, no medical and demographic information, no spirometry, no lab tests) or FEV1/FVC ratio >0.70 after administration of bronchodilator.

### Smoking status

Every participant was classified according to smoking status (*Schoenborn & Adams, 2010*):

Never smoker—never smoked a cigarette or who smoked fewer than 100 cigarettes in their entire lifetime.

Former smoker—smoked at least 100 cigarettes in their entire life but were not currently smoking.

Current smoker—had smoked at least 100 cigarettes in their entire life and were still smoking.

Numbers of pack-years were calculated using the formula:

Number of pack-years = years of smoking × number of daily smoked cigarettes/20.

## Anthropometric indices

Body weight and height were measured and the body mass index (BMI) was calculated by dividing weight by height squared (kg/m$^2$). According to BMI all patients were classified as underweight (<18.5), normal (18.5–24.99), overweight (25–29.99) and obese (>30). Waist circumference was measured at the approximate midpoint between the lower margin of the last palpable rib and the top of the iliac crest according to the WHO STEPS protocol (*WHO, 2008*). Hip circumference was measured around the widest portion of the buttocks (*WHO, 2008*). Body adiposity index (BAI) was calculated as:

Hip circumference (in cm)/ (Height (in m) X$\sqrt{}$Height) − 18.

## COPD exacerbations and duration of hospital stay

Data were gathered for number of severe exacerbations (hospitalizations) and moderate exacerbations (antibiotic or/and systemic steroid treatment without hospitalization due to worsening of pulmonary symptoms) (*Global Initiative for Chronic Obstructive Lung Disease , 2014*) in the previous year. The duration of the current hospital stay was recorded.

## Quality of life

Quality of life was assessed with the mMRC scale and CAT questionnaire. Patients were instructed that there were no right or wrong answers. All patients' questions were answered. Patients were classified according to GOLD as having less symptoms (CAT < 10) and breathlessness (mMRC grade 0–1) and more symptoms (CAT $\geq$ 10) and breathlessness (mMRC grade $\geq$ 2). Because all patients were hospitalized due to exacerbation there were only group C (high risk, less symptoms) and group D (high risk, more symptoms) patients according to GOLD (*Global Initiative for Chronic Obstructive Lung Disease , 2014*).

## Pulmonary function testing

The spirometry was performed using Minispir$^{\circledR}$ New spirometer (MIR—Medical International Research, Rome, Italy). Patients were instructed to withdraw using short-acting $\beta$2-agonists at least 6 h, long-acting $\beta$2-agonist at least 12 h, long acting muscarinic antagonist 24 h and short acting muscarinic antagonist 12 h before the spirometry (*Miller et al., 2005*). Post bronchodilator spirometry testing was performed 15–30 min after inhalation of 400mcg Salbutamol according to ERS/ATS recommendations (*Miller et al., 2005*). Pre- and post- values were obtained for: FVC, FEV1, FEV1/FVC, FEV6, FEV1/FEV6, PEF, FEF2575, FEV3, FEV3/FVC as well as the difference between post/pre values (delta values). GLI (Global Lungs Initiative) predicted values were used (GLI-2012). Patient's obstruction was classified according to the severity of airflow limitation based on post-bronchodilator FEV1 as follows: mild ($\geq$80% predicted); moderate (80>FEV1 $\geq$ 50% predicted); severe (50%>FEV1$\geq$30% predicted); very severe (<30% predicted) (*Global Initiative for Chronic Obstructive Lung Disease , 2014*).

## Blood samples and analyses

A venous blood sample was collected from each subject after a 12-h fasting. Blood samples were taken as late as possible before discharging (usually on 6th or 7th day). Plasma glucose, triglyceride (TG), high density lipoprotein (HDL), low density lipoprotein (LDL), and total cholesterol (tChol) were measured with a Roche COBAS INTEGRA® 400 plus analyzer and an enzymatic colorimetric assay and blood glucose was measured with an enzymatic reference method with hexokinase. Vitamin D was measured with Elecsys 2010 (Roche, Basel, Switzerland) and Electro-chemiluminescence immunoassay (ECLIA). Glycated hemoglobin (HbA1c) was measured with a NycoCard device and boronate affinity assay. For patients without established DM a 75 g OGTT was performed with blood samples for glucose taken on first and second hour.

## Statistical analysis

Statistical analysis was performed with the SPSS for Windows software, version 22.0 (SPSS Inc., Chicago, Illinois, USA). Continuous variables were presented as mean ± standard deviation and 95 Confidence intervals (95%CI) and categorical variables—as percentages. Chi-square test was used to determine the associations between categorical variables. Continuous variables were examined for normality by Shapiro–Wilk test. For normally distributed variables, differences between the groups were determined by independent-samples $T$ test for two samples and analysis of variance (ANOVA) for more than 2 samples. Mann–Whitney U test was used for abnormally distributed variables with 2 samples and Kruskal-Wallis test for variables with more than 2 samples. Regression analyses were used to determine risk factors for presence of MS or the consequences of having MS. Significance value ($p$-value) was set at 0.05.

All patients signed informed consent. Medical University-Sofia Research Ethics Commission approved the study (#2976/2014).

# RESULTS

## Sample characteristics

A total of 152 COPD patients admitted for exacerbation were recruited from University Specialized Hospital for Active Treatment of Pulmonary Diseases 'Saint Sofia,' Sofia, Bulgaria. Mean age of patients in this study was 65 ± 10 years. 71.1% (108/152) were males, 28.9% (44/152) were females; mean post-bronchodilator $FEV_1$ was 55.3 ± 19.5%. 15.8% from the patients were never smokers, 57.9%—former smokers and 26.3%—current smokers. 127 patients (83.6%) were receiving inhalatory corticosteroids.

## Prevalence of MS

25.0% (38/152) of the patients have MS. 23.1% (25/108) of males have MS vs. 29.5% (13/44) of females but this difference is not statistically significant (Table 1). Mean age does not differ between the patients with and without MS.

Fulfilled criteria for MS (in all and in MS patients) are shown in Table 2. Virtually all (37/38) patients with MS in this study have arterial hypertension, followed by elevated

**Table 1 Prevalence of MS according to different factors.**

| | % MS | P value |
|---|---|---|
| **All** | **25.0** | |
| **Sex** | | |
| Male | 23.1 | |
| Female | 29.5 | P = 0.409 |
| **Smoking status** | | |
| Never | 25.0 | |
| Former | 27.3 | P = 0.678 |
| Current | 20.0 | |
| **ICS use** | | |
| Yes | 25.2 | |
| No | 24.0 | P = 0.899 |
| **Arterial hypertension** | | |
| Yes | 35.2 | |
| No | 2.1 | P<0.0005 |
| **Vitamin D status** | | |
| >50 nmol/l | 24.2 | |
| 25–50 nmol/l | 24.6 | P = 0.929 |
| <25 nmol/l | 28.0 | |
| **DM** | | |
| Yes | 37.7 | |
| No | 18.2 | P=0.008 |
| **BMI** | | |
| Underweight | 0 | |
| Normal | 0 | P<0.0005 |
| Overweight | 27.3 | |
| Obese | 54.8 | |
| **BAI** | | |
| Underweight | 0 | |
| Normal | 17.5 | |
| Overweight | 28.9 | P = 0.001 |
| Obese | 50.0 | |
| **Quality of life** | | |
| CAT 0–9 | 16.0 | |
| CAT ≥ 10 | 26.8 | P = 0.256 |
| mMRC 0 or 1 | 18.9 | |
| mMRC ≥ 2 | 28.3 | P= 0.201 |
| **FEV1** | | |
| FEV1≥50% | 21.3 | |
| FEV1<50% | 27.5 | P = 0.390 |
| FEV1≥80% | 11.8 | |
| 80%>FEV1≥50% | 28.8 | |
| 50%>FEV1≥30% | 26.1 | P = 0.852 |
| FEV1<30% | 25.0 | |

**Table 2  Fulfilled criteria for MS.**

| MS criteria | All patients ($n = 152$) | MS only ($n = 38$) | Without MS ($n = 114$) | Accuracy |
|---|---|---|---|---|
| Elevated blood pressure: systolic ≥130 and/or diastolic ≥85 mm Hg (or on therapy) | 69.1% ($n = 105$) | 97.4% ($n = 37$) | 59.6% ($n = 68$) | 54.6% |
| Elevated waist circumference >102 cm in males, >88 cm in females | 28.3% ($n = 43$) | 86.8% ($n = 33$) | 8.8% ($n = 10$) | 90.1% |
| Triglycerides >1.7 mmol/L (or on therapy) | 29.6% ($n = 45$) | 60.5% ($n = 23$) | 19.3% ($n = 22$) | 75.7% |
| Fasting glucose >5,5 mmol/L (or on therapy) | 34.2% ($n = 52$) | 65.8% ($n = 25$) | 23.7% ($n = 27$) | 73.7% |
| HDL <1.0 mmol/L in males, <1.3 mmol/L in females (or on therapy) | 15.8% ($n = 24$) | 39.5% ($n = 15$) | 7.9% ($n = 9$) | 78.9% |

**Table 3  Number of fulfilled criteria for MS in all patients.**

| Number of fulfilled criteria | % | n |
|---|---|---|
| 0 | 13.8 | 21 |
| 1 | 33.6 | 51 |
| 2 | 27.6 | 42 |
| 3 | 13.2 | 20 |
| 4 | 10.5 | 16 |
| 5 | 1.3 | 2 |

waist circumference (33/38). Arterial hypertension has greatest sensitivity for predicting presence of MS, but is not specific. Elevated waist circumference has greatest accuracy, defined as sum of true results (true positives and true negatives) divided by total number of cases. Number of fulfilled criteria in all patients is given in Table 3.

## Lifestyle factors

Our study did not find significant differences in prevalence of MS according to smoking status and number of pack-years. Treatment with inhalatory corticosteroids (ICS) is not risk factor for MS. Fasting glucose level are not influenced by ICS (all $p > 0.05$) (Table 1).

## Comorbidity

In our study, vitamin D levels do not significantly differ in relation to presence of MS (32.11 vs. 31.92 nmol/l). The presence of MS is also not related to vitamin D status (Table 1).

Number of fractures in our study does not significantly differ regarding the presence of MS. Presence of at least one fracture also does not differ significantly in relation to presence of MS.

In our study there is a correlation between the presence of MS and DM. 52.6% (20/38) from patients with MS have DM. BMI and BAI differ significantly according to presence of MS—32.51 vs. 25.58, for BMI and 31.01 vs. 25.58, for BAI. There is also significant difference in prevalence of MS in BMI and BAI groups (Table 1). It is notable that none of the patients have MS and BMI < 25 (Table 1).

**Table 4  Number of exacerbations in previous year and duration of hospital stay.**

|  | No MS | MS |
|---|---|---|
| Moderate exacerbations | 0,61 (0,49–0,76) | 0,92 (0,59–1,34) |
| Severe exacerbations | 1,79 (1,61–1,97) | 2,08 (1,71–2,50) |
| All exacerbations | **2,40 (2,19–2,61)** | **3,00 (2,56–3,52)** |
| Hospital stay (in days) | 7,47 (7,24–7,70) | 7,63 (7,28–8,05) |

Linear regression showed presence of MS as risk factor for higher BMI ($R = 0.542$, $r^2 = 0.293$, $p < 0.0005$, B $= 6.928$, 95% CI [5.193–8.662]) and BAI ($R = 0.406$, $r^2 = 0.165$, $p < 0.0005$, B $= 5.423$, 95% CI [3.455–7.392]).

A logistic regression analysis was conducted to predict presence of MS in a relation to presence of other comorbidities. Presence of DM slightly improves the model (chi square $= 6.818$, $p = 0.009$ with $df = 1$). Nagelkerke's R2 of 0.065 indicates a weak relationship. Odds ratio was 2.73. Vitamin D status does not improve the model ($p > 0.05$).

## Exacerbations and duration of hospital stay

Our study found a significant difference between the number of total exacerbations according to the presence of MS (Table 4, $p = 0.015$). The number of severe exacerbation, moderate exacerbation and duration of hospital stay did not reach significance.

Triglycerides and blood glucose levels in our study did not correlate with number of exacerbations.

Linear regression showed presence of MS as risk factor for higher number of exacerbations ($R = 0.207$, $r^2 = 0.043$, $p = 0.010$, B $= 0.596$, 95% CI [0.143–1.050]). From the MS components presence of arterial hypertension is strongest risk factor for exacerbation (R $= 0.228$, $r^2 = 0.052$, $p = 0.005$, $B = 0.615$, 95% CI [0.192–1.038]).

## Quality of life

The presence of MS is associated with significantly worse cough and sleep (1st and 7th CAT questions; $p = 0.002$ and $p = 0.001$ respectively) and higher total CAT score ($p = 0.017$) (Table 5). However prevalence of MS is not significantly different between patients with less symptoms (CAT 0–9) and breathlessness (mMRC 0 or 1) compared to patients with more symptoms (CAT $\geq 10$) and breathlessness (mMRC $\geq 2$) (Table 1).

Regression analyses also showed that MS is a risk factor for reduced quality of life, measured with total CAT score ($R = 0.205$, $r^2 = 0.042$, $p = 0.011$, B $= 3.711$, 95% CI [0.859–6.562]). Presence of MS also impairs cough and sleep—first (R $= 0.285$, $r^2 = 0.081$, $p < 0.0005$, B $= 0.684$, 95% CI [0.313–1.055]) and seventh (R $= 0.268$, $r^2 = 0.072$, $p = 0.001$, $B = 0.930$, 95% CI [0.390–1.470]) CAT questions.

## Pulmonary function test (PFT)

Our study did not find differences in FVC, FEV1, FEV1/FVC, FEV6, FEV1/FEV6, PEF, FEF2575 and FEV3 according to the presence of MS. It should be noted that there is tendency for FVC and FEV1/FVC ratio. However, because of this there is significant difference in FEV3/FVC ratio (Table 6).

**Table 5  Mean CAT score on every question and in total according to presence of MS.**

| MS | Mean CAT score | N | P value |
|---|---|---|---|
| MS—no | CAT1 1.95 | 114 | **P=0.002** |
| MS—yes | CAT1 2.63 | 38 | |
| MS—no | CAT2 1.92 | 114 | $P = 0.063$ |
| MS—yes | CAT2 2.34 | 38 | |
| MS—no | CAT3 2.54 | 114 | $P = 0.092$ |
| MS—yes | CAT3 2.97 | 38 | |
| MS—no | CAT4 3.52 | 114 | $P = 0.361$ |
| MS—yes | CAT4 3.74 | 38 | |
| MS—no | CAT5 1.23 | 114 | $P = 0.198$ |
| MS—yes | CAT5 1.66 | 38 | |
| MS—no | CAT6 1.54 | 114 | $P = 0.695$ |
| MS—yes | CAT6 1.68 | 38 | |
| MS—no | CAT7 1.28 | 114 | **P=0.001** |
| MS—yes | CAT7 2.21 | 38 | |
| MS—no | CAT8 2.62 | 114 | $P = 0.068$ |
| MS—yes | CAT8 3.08 | 38 | |
| MS—no | Total CAT 16.61 | 114 | **P = 0.017** |
| MS—yes | Total CAT 20.32 | 38 | |

**Table 6  Mean PFT values.**

| MS | Mean PFT value | N | P value |
|---|---|---|---|
| No | FEV1 55.56% | 114 | $P = 0.811$ |
| Yes | FEV1 54.68% | 38 | |
| No | FVC 80.46% | 114 | $P = 0.094$ |
| Yes | FVC 72.45% | 38 | |
| No | FEV1/FVC 0.53 | 114 | $P = 0.091$ |
| Yes | FEV1/FVC 0.57 | 38 | |
| No | FEV6 73.89% | 114 | $P = 0.277$ |
| Yes | FEV6 68.63% | 38 | |
| No | FEV1/FEV6 0.57 | 114 | $P = 0.107$ |
| Yes | FEV1/FEV6 0.61 | 38 | |
| No | PEF 55.62% | 114 | $P = 0.735$ |
| Yes | PEF 56.66% | 38 | |
| No | FEF2575 38.89% | 114 | $P = 0.316$ |
| Yes | FEF2575 40.95% | 38 | |
| No | FEV3 66.62% | 114 | $P = 0.601$ |
| Yes | FEV3 63.89% | 38 | |
| No | FEV3/FVC 0.81 | 114 | **P =0.033** |
| Yes | FEV3/FVC 0.85 | 38 | |

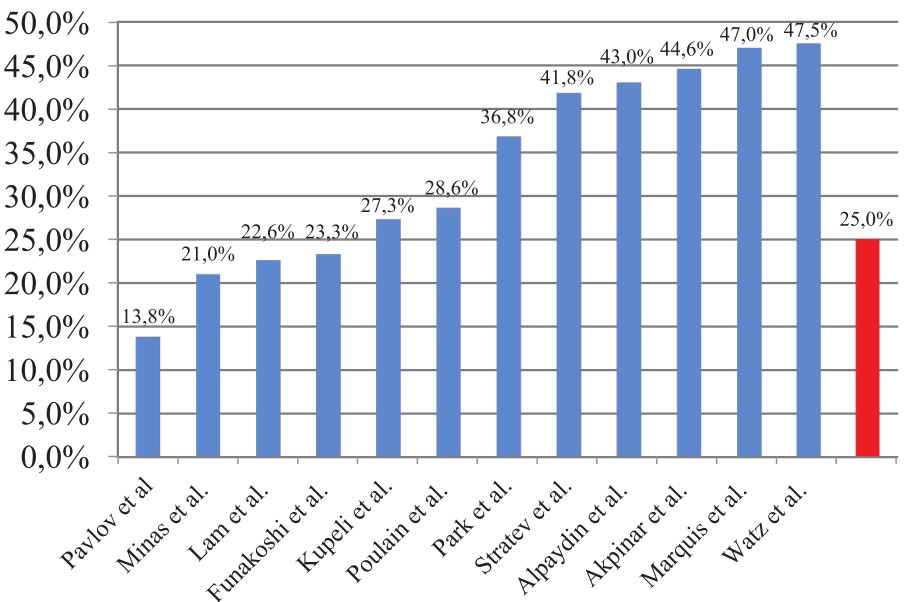

**Figure 1 Prevalence of MS in COPD patients.**

Regression analyses also showed that MS is not a risk factor for reduced pulmonary function. However some of the components of MS are associated with reduced pulmonary function with highest impact of HDL on FVC ($R = 0.183$, $r^2 = 0.033$, $p = 0.024$, $B = -8.517$, 95% CI [$-15.904$–$-1.130$]) and FEV1 ($R = 0.251$, $r^2 = 0.063$, $p = 0.001$, $B = -10.391$, 95% CI [$-16.865$–$-3.918$]). Fasting glucose is associated with increased FEV1/FVC ratio ($R = 0.186$, $r^2 = 0.035$, $p = 0.022$, $B = 1.238$, 95% CI [0.183–2.294]) probably because of lowering FVC.

There is no difference in prevalence of MS in patients with FEV1 <50%, when compared to patients with FEV1 >50% or regarding GOLD stage (Table 1).

## DISCUSSION

This study found comparable but relatively low prevalence of MS compared to previous studies (Table 7) (Fig. 1). The prevalence of MS in our study is significantly lower when compared to the general Bulgarian population (44.6% in subjects over 45 years) (*Borissova et al., 2007*). The odds ratio for COPD patients admitted for exacerbation of having MS is 0.41 compared to general population (95% CI [0.28–0.61], $p < 0.0005$).

Two Bulgarian studies examined the prevalence of MS in COPD patients. An epidemiological study conducted in Bulgaria in COPD patients reported a prevalence of 13.8%. These results differ significantly from the literature data probably because of the different criteria for metabolic syndrome (presence of DM, BMI >30 and blood pressure >140/90 mmHg) which makes data comparing irrelevant (*Pavlov et al., 2010*). A more recent study indicates that the prevalence of metabolic syndrome is 41.8% in patients with COPD . However, this study was not conducted using a random sample (exclusion criteria was presence of DM, people were aged 49–79 years for patients with COPD and

**Table 7 Prevalence of MS in patients with COPD.**

| Authors | N | Studied population | Prevalence of MS |
|---|---|---|---|
| *Akpinar et al., 2012* | 133 | Patients with COPD and controls | **44.6%** |
| *Funakoshi et al., 2010* | 7.189 | Men aged 45–88 years | **16.8%**, OR 0.72 (95% CI [0.51–1.02]) in GOLD I; **28.7%**, OR 1.33 (95% CI [1.01–1.76]) in GOLD II–IV |
| *Kupeli et al., 2010* | 106 | Hospitalized patients with COPD | **27.3%** |
| *Lam et al., 2010* | 7.358 | General population >50 years | **22.6%**; OR 1.47 (95% CI [1.12–1.92]) |
| *Marquis et al., 2005* | 72 | Patients with COPD and controls | **47%** |
| *Minas et al., 2011* | 114 | Men with COPD | **21%** |
| *Ozgen Alpaydin et al., 2013* | 90 | Patients with COPD and controls | **43%** |
| *Park et al., 2012* | 1.215 | Patients with COPD and controls >40 years | **33%** vs. 22.2% for men; **48.5%** vs. 29.6% for women OR 2.03 (95% CI [1.08–3.80]) |
| *Poulain et al., 2008* | 28 | Patients with COPD | Overweight—**50%**; Normal weight—**0%**. |
| *Watz et al., 2009* | 200 | Patients with COPD and chronic bronchitis | GOLD I—**50%**; GOLD II—**53%**; GOLD III—**37%**; GOLD IV—**44%**; Chronic bronchitis—53% |

35–65 years for controls) (*Stratev et al., 2012*). Nonetheless it uses similar criteria for MS and when comparing the results our study finds lower prevalence of MS.

The prevalence results could be explained with differences between the populations in different studies (physical activity, diet, lifestyle etc.). For example, Bulgaria is low-income country, which may impact diet preferences and treatment choices. Furthermore, patients in this study had been hospitalized due to exacerbation, which represents the most severe group of COPD patients. Having in mind that MS is more common in the early stages and decreases with COPD progression (*Watz et al., 2009*), COPD patients hospitalized for exacerbation may be considered as having advanced COPD.

According to the NHANES III study, smokers are more likely to develop MS than nonsmokers in general population, and the risk increases with the number of pack-years even after adjusting for covariates (*Park et al., 2003*). Our study did not find significant differences in prevalence of MS according to smoking status and number of pack-years. These results could be explained with smoke being the biggest factor in developing COPD and effect of developing MS could be reduced. Moreover nicotine may be an appetite suppressant and lower the weight thus decreasing prevalence of metabolic syndrome (*Chiolero et al., 2008*). Third, hospitalized COPD patients are patients with predominantly advanced disease and prone to cachexia and wasting. Also, lifestyle changes (quiting smoking) in the presence of the two diseases should be considered which may change the prevalence of MS.

Treatment with inhalatory corticosteroids (ICS) is not risk factor for MS similar to findings for DM (*O'Byrne et al., 2012*) and fasting glucose level are not influenced by ICS.

COPD is a disease that affects mainly the lungs, but is characterized by systemic inflammation and a number of extrapulmonary manifestations. Only 1/3 of patients with COPD die due to respiratory failure. Main cause of death is lung cancer and cardiovascular complications (*Calverley et al., 2007*).

The vast majority of patients with COPD have a vitamin D deficiency (*Romme et al., 2013*). Aside from its role in the metabolism of calcium and phosphorus, vitamin D is involved in the pathogenesis of multiple diseases, including MS, mainly because it affects the secretion and the function of insulin (*Ju, Jeong & Kim, 2014*). However, in our study vitamin D levels do not significantly differ in relation to presence of MS. Presence of MS is also not related to vitamin D status.

There are no studies that examine the relationship between osteoporosis and MS in patients with COPD. However, both diseases share common risk factors such as smoking, lack of physical activity, and treatment with corticosteroids. Some of the components of the metabolic syndrome (arterial hypertension, elevated triglycerides, reduced HDL cholesterol) are risk factors for low bone density. Systemic inflammation in MS plays a role in the pathogenesis of osteoporosis (*McFarlane, 2006*). On the other hand, studies examining the relationship between MS and osteoporosis showed inconsistent results, probably due to the protective effect of obesity (*Zhou et al., 2013*). However, the number of fractures in our study does not significantly differ regarding the presence of MS. The presence of at least one fracture also does not differ significantly in relation to presence of MS.

Most patients with DM have MS, but the opposite is not necessarily true (*Ginsberg & Stalenhoef, 2003*). The presence of MS in this study is associated with presence of DM, higher BMI and BAI.

Hyperglycemia is associated with elevated glucose concentrations in tissues and bronchial aspirates where it may stimulate infection by enhancing bacterial growth and by promoting bacterial interaction with the airway epithelium (*Brennan et al., 2007*). Hyperglycemia also impairs both innate and adaptive immunity, suppressing the host response to infection.

The presence of MS in patients with COPD increases the frequency of exacerbations (2.4 vs. 0.7) and their duration–(7.5 vs. 5.0 days) according to *Kupeli et al. (2010)*, and 8 versus 5.5 days, according to *Abdelghaffar et al. (2012)*. Our study found a significant difference between the number of total exacerbations according to the presence of MS. However, the number of severe exacerbation, moderate exacerbation and duration of hospital stay did not differ significantly. Triglycerides and blood glucose levels in our study did not correlate with number of exacerbations as reported by other authors (*Kupeli et al., 2010*).

The presence of MS is associated with significantly worse cough and sleep and higher total CAT score. This confirms the data about reduced quality of life in patients with MS (*Ford & Li, 2008*). However, the prevalence of MS is not significantly different between patients with less symptoms (CAT 0–9) and breathlessness (mMRC 0 or 1) compared to patients with more symptoms (CAT $\geq$10) and breathlessness (mMRC $\geq$2). These mixed results may be explained with COPD having higher negative impact on quality of life than MS (physical limitation due to shortness of breath) as suggested for DM (*Arne, Janson & Janson, 2009*), and ameliorating the effect in patients having both diseases.

COPD is characterized by airflow obstruction, which is not fully reversible. MS is associated with a reduction of lung volumes (*Fimognari et al., 2007*; *Leone et al., 2009*; *Lin et al., 2006*; *Nakajima et al., 2008*; *Yeh et al., 2011*). It should be noted that some studies

found no association between lung function and the presence of MS (*Yamamoto et al., 2014*). MS in our study is not associated with worsen pulmonary function. There is also no difference in prevalence of MS in patients with FEV1 <50%, when compared to patients with FEV1 >50% or regarding GOLD stage.

## CONCLUSIONS

This study finds a 25% prevalence of MS in COPD patients admitted for exacerbation, which is significantly lower than the general population. MS is more prevalent in females, but the gender difference is not statistically significant. In this study, most of the patients are former smokers, and the prevalence of MS does not differ regarding smoking status and treatment with ICS.

The presence of MS is associated with the presence of DM, higher BMI and BAI, more exacerbations during the previous year and lower quality of life. MS is not associated with increased hospital stay and lower pulmonary function.

This study finds comparable but relatively low prevalence of MS compared to previously published data (21–53%). As MS is more common in the early stages and decreases with COPD progression, the COPD patients admitted for exacerbation may be considered as having advanced COPD.

### Funding

This manuscript is part of a PhD project, which is partially funded by Medical University—Sofia, Sofia, Bulgaria (grant number 15-D/2014, project number 22-D/2014). The funders had no role in study design, data collection and analysis, decision to publish, or preparation of the manuscript.

### Grant Disclosures

The following grant information was disclosed by the authors:
Medical University—Sofia: 15-D/2014.

### Competing Interests

The authors declare there are no competing interests.

### Author Contributions

- Evgeni Mekov conceived and designed the experiments, performed the experiments, analyzed the data, contributed reagents/materials/analysis tools, wrote the paper, prepared figures and/or tables, reviewed drafts of the paper.
- Yanina Slavova conceived and designed the experiments, contributed reagents/materials/analysis tools, wrote the paper, reviewed drafts of the paper.
- Adelina Tsakova and Marianka Genova performed the experiments, contributed reagents/materials/analysis tools, reviewed drafts of the paper.

- Dimitar Kostadinov and Delcho Minchev contributed reagents/materials/analysis tools, reviewed drafts of the paper.
- Dora Marinova contributed reagents/materials/analysis tools, wrote the paper, reviewed drafts of the paper.

## Human Ethics

The following information was supplied relating to ethical approvals (i.e., approving body and any reference numbers):

Medical University-Sofia Research Ethics Commission approved the study (#2976/2014).

## Data Deposition

The following information was supplied regarding the deposition of related data:

http://figshare.com/articles/MS_in_COPD/1439301.

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
