# Peer review of "Metabolic syndrome in hospitalized patients with chronic obstructive pulmonary disease"

_PeerJ, doi:10.7717/peerj.1068_

## Round 0.1 · original submission · Major Revisions

My review

In this study Mekov et al. investigated the prevalence of MS and its correlation with comorbidities in 152 COPD patients admitted for exacerbation. Although facing an interesting topic and the Authors performed a lot of work for data analysis, a number of flaws limits the potential interest of the study.

Major issues are the design and sample size. The study, which is retrospective, used a large amount of data collected in 152 patients during their hospital stay. A clear design, e.g. establishing the time period during which patients were observed, is lacking. Moreover, being a prevalence study, size should be accordingly dimensioned.

Another issue is that MS entails a syndrome that includes patients at high cardiovascular risk for the presence of a cluster of factors that individually implies limited risk. Exacerbation of COPD is an acute condition that may affect metabolism, possibly resulting in an over- or under-estimation of MS. A further point could be to understand if estimation of MS during exacerbation could be useful for assessing the cardiovascular risk.
Since COPD exacerbation is an acute and critical condition, and diabetes may be overt when frailness and other morbidities coexist, appropriate comparisons could be made with studies enrolling acute patients.

Specific comments

Data are presented in a confusing way, with overlap between previous data of the literature and original data. I strongly suggest a re-arrangement of the entire manuscript giving emphasis to the novel data. Moreover, values should not been reported in both Tables and text, providing redundant duplicates.
Some sentences should be rephrased. As an example, in the abstract: MS may also correlate with disease characteristics. The meaning of the statement could be more clearly explained.
There are too many references for this type of manuscript.

I suggest to carefully take into consideration also the criticisms made by the Reviewer 1 before revising the manuscript. As you can see, some criticisms on methodology and presentation of the results were raised independently by both reviewers.

·

Basic reporting

This is an interesting study evaluating the COPD and metabolic syndrome. Background and prior literature are properly presented, but right from the introduction the authors are "jumping" to the conclusions of their study, justifing unexpected results.

Experimental design

The study was well designed , and the authors show good and expertise knowledge in COPD. The authors are critically taking their results, but I think that concernig metabolic syndrome (MS) they are missing the proper cardiovascular background. MS, as far as I can argue, is taken more as a definition than as a clinical predisposition to develop cardiovascular disease. Patients included in the study mostly had already hypertension, DM or were obese.
Utility to use BAI in addition to BMI is not explained
The evaluation of Vit.D deficiency and COPD, in my opinion, is not fulfilling the aims of the study.
Statistical analysis is properly applied, however not always necessary (correlation between BMI and MS, DM and MS).
Conclusions are too much summerized and are not giving the consistent importance to the results: I would suggest to present results apart from the discussion.

Validity of the findings

Results are really interesting but data about percentage of patients in therapy for COPD, as well as which therapy (short and long acting drugs) are missing. These data are necessary for the pubblication, as well as to explain, at least, part of the results.

Additional comments

The results are interesting, and I think a different structure of the paper would give to it the proper relevance.

---

## Round 0.2 · Minor Revisions

I appreciated the efforts made by the Authors to improve the manuscript. However, it needs to be further improved before being considered for publication.

I suggest some changes, as detailed below. The most crucial issue is redundancy of the text and replica of data.

Specific points
Page 2, line 10: … higher – 44.6%. Hyphen is misleading because it could be read as minus. I suggest to replace – with , i.e. (or, alternatively with approximately 44%).
Page 2, line 11: I suggest the following change: … in 3598 COPD patients instead of COPD patients (n= 3598) . Similar changes could be made in the next sentence.
Page 3. Units could be indicated in BAI formula.
Page 4, line 13: Patients’ obstruction were … should be replaced with Patient’s obstruction was …(not were)
Page 5. Please remove decimals from age values. E.g. 65 not 65.1.
Use only one decimal for FEV1 values.
When p is not significant please omit p value in the text (also for the other parameters).
Please, avoid duplications of p and r values in the text and tables.
Page 5. What is the meaning of the sentences ‘Arterial hypertension has greatest sensitivity but is not specific. Elevated waist circumference has greatest accuracy’ in the paragraph Results. Prevalence of MS?

Titles of paragraphs in Results Section. Please omit ‘results’ because you are reporting results in the Results Section. Hence: replace Comorbidity results with Comorbidity, Exacerbation results with Exacerbation, Quality of life results with Quality of life, Pulmonary function test results with Pulmonary function test.

Avoid replica of data in the Discussion if data are reported in Results. Likewise, do not report details of the literature if reported in the Introduction. Why did you report the number of patients (n= 3598) enrolled in the Bulgarian study in both Introduction and Discussion? The same is true for Watz’s study (reference 24).
Most data reported in the first paragraph of Discussion were previously reported in Introduction. Remove duplicates leaving only comments on differences between previous studies and the present one.
Replica of information makes difficult to understand the novelty of your data. Please make smooth and easy the reading of your manuscript.

Tables must be numbered. A brief title is only needed.

My opinion is that there are still too many references.

---

## Round 0.3 · accepted · Accept

I appreciated the efforts made by the Authors to improve the manuscript.